# SEGO: Sequential Subgoal Optimization for Mathematical Problem-Solving

## Abstract

Large Language Models (LLMs) have driven substantial progress in artificial intelligence in recent years, exhibiting impressive capabilities across a wide range of tasks, including mathematical problem-solving. Inspired by the success of subgoal-based methods, we propose a novel framework called **SE**quential sub**G**oal **O**ptimization (SEGO) to enhance LLMs' ability to solve mathematical problems. By establishing a connection between the subgoal breakdown process and the probability of solving problems, SEGO aims to identify better subgoals with theoretical guarantees. Addressing the challenge of identifying suitable subgoals in a large solution space, our framework generates problem-specific subgoals and adjusts them according to carefully designed criteria. Incorporating these optimized subgoals into the policy model training leads to significant improvements in problem-solving performance. We validate SEGO's efficacy through experiments on two benchmarks, GSM8K and MATH, where our approach outperforms existing methods, highlighting the potential of SEGO in AI-driven mathematical problem-solving. [1]

## 1 Introduction

In recent years, the emergence of Large Language Models (LLMs) has marked a significant milestone in the field of artificial intelligence. Models such as ChatGPT and LLaMA have demonstrated remarkable capabilities across diverse tasks. Within this context, addressing mathematical problems has attracted considerable interest from researchers, as it serves as a prominent showcase of the reasoning capabilities inherent in LLMs. Reasoning involves a multitude of aspects, among which the ability to decompose the overall problem into smaller, more manageable subproblems (i.e., subgoals) is particularly essential for effective problem-solving.

In this paper, we draw inspiration from the successful application of subgoal-based methods in both RL and LLMs (Zhang et al., 2020; Zhao et al., 2023) and introduce a novel framework called SEGO (SEquential subGoal Optimization). Intuitively, a good subgoal should serve as a bridge to solving a bigger problem, such that breaking down the problem into these subgoals makes the subproblems easier to solve, thereby increasing the likelihood of solving the entire problem. SEGO quantifies this intuition by establishing a theoretical connection between the subgoal breakdown process and the probability of solving the problem (Eq. 1). Concretely, we construct a lower bound on the probability of solving the complete problem using a proposal distribution considering a specific subgoal. We then employ a method inspired by annealed importance sampling (Neal, 2001) to efficiently navigate through vast search spaces, seeking the subgoal corresponding to the theoretically optimal proposal distribution, while ensuring the process doesn't get trapped in suboptimal subgoals (§2.3). By incorporating these sequentially optimized subgoals into the training of the policy model, we achieve significant improvements in solving mathematical problems.

To empirically validate the efficacy of SEGO, we conducted experiments on two primary benchmarks: GSM8K and MATH. Our approach demonstrated marked superiority against existing methods with comparable model sizes, highlighting the potential of SEGO in advancing the field of AI-driven mathematical problem-solving. We hope that our findings can open up new avenues for future research on the applicability of LLMs to complex tasks in diverse domains (Yao et al., 2022; Liu et al., 2023).

---

[1] All data and code will be open-sourced upon publication.

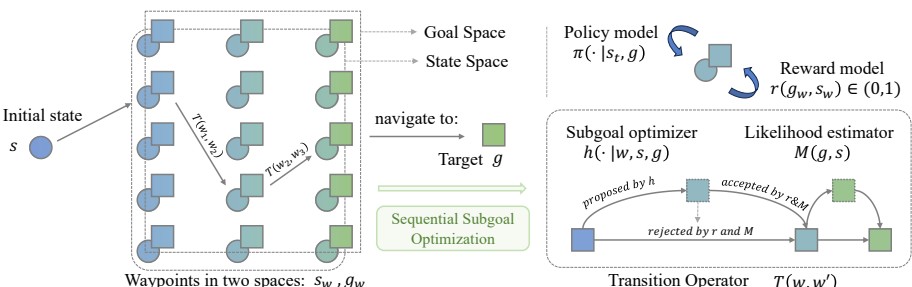

Figure 1: An overview of the SEGO framework, which uses waypoints to traverse the mathematical problem-solving space. Starting from an initial state $s_0$, SEGO proposes a "draft" waypoint (i.e., subgoal) $\omega_1$ and optimizes it through a sequential subgoal optimization process. This process integrates the feedback from the likelihood estimator, determining the probability of reaching a goal $g$ from a state $s$. A reward model is used to evaluate whether a goal has been achieved in a state by measuring their proximity.

## 2 METHODOLOGY

### 2.1 OVERVIEW

In this section, we introduce SEGO, illustrated in Figure 1, a novel framework that synergizes the principles of goal-conditioned Reinforcement Learning with mathematical problem-solving. Within this framework, an *action* represents a step in the solution, marking a logical progression towards resolving a problem. The *state*, which summarizes the current progress, is the combination of the actions taken. The *(sub-)goal* refers to the intended mathematical (sub-)problem to be solved, and the *trajectory* describes the sequences of actions executed to reach this goal. SEGO primarily focuses on the generation and sequential optimization of a subgoal which serves as a bridge to navigating more intricate challenges. These trajectories, connecting initial states to the subgoal and subsequently to the final goal, are then used to optimize our policy model, enhancing its proficiency in resolving sophisticated problems.

**Road Map** We first introduce the learning objectives and intricate optimization challenges in §2.2. The workflow and strategic design of SEGO, along with the introduction of auxiliary modules and their roles in sequential subgoal sampling and navigating through complex problem-solving spaces, are illustrated in §2.3.

### 2.2 LEARNING OBJECTIVE AND CHALLENGES

Expanding on the foundational concepts introduced, the goal-conditioned RL framework of SEGO is designed to optimize the problem-solving process. An agent begins in an initial state $s$, representing the starting point in solving the problem. As the agent selects actions, each representing a logical step toward the solution, it progresses through the state space, aiming to resolve the specified mathematical problem or goal $g$. The binary reward function $r(g, s)$ is defined as 1 if the goal $g$ is reached at the current state $s$, and 0 otherwise. The probability of reaching a goal $g$ from a state $s$ under policy $\pi(\cdot|\cdot, g)$ is represented as $p^{\pi(\cdot|\cdot,g)}(g|s)$. The task is to find the best policy that can maximize this probability:

$$\pi^\star = \arg\max_\pi p^{\pi(\cdot|\cdot,g)}(g \mid s).$$

We further enhance our approach to adeptly navigate through the intricacies of mathematical problem-solving in the SEGO framework. Specifically, we employ a strategy that utilizes waypoints to traverse the state space effectively, acting as bridges to solve larger problems. The evidence lower bound ($\mathcal{L}$) is central to this methodology, serving as a metric to quantify the agent's probability of successfully reaching the target and establishing a theoretical connection between the subgoal breakdown process and the probability of solving the problem:

**Proposition 2.1.** *The objective $\mathcal{L}$, defined below, constitutes a lower bound on the probability of reaching the goal $g$ from state $s$:*

$$\log p^{\pi(\cdot|\cdot,g)}(g \mid s) \geq \mathbb{E}_{q(g_w, s_w|g,s)}\bigg[ \log p^{\pi(\cdot|\cdot,g)}(g \mid s_w) + \log p^{\pi(\cdot|\cdot,g_w)}(g_w \mid s) + r(g_w, s_w) \\ - \log q(g_w, s_w \mid g, s)\bigg] \triangleq \mathcal{L}. \tag{1}$$

We provide the proof in Appendix A.1. The term $p^{\pi(\cdot|\cdot,g)}(g \mid s_w)$ represents the transition probability from an *intermediate state* $s_w$ to the ultimate goal. Similarly, $p^{\pi(\cdot|\cdot,g_w)}(g_w \mid s)$ assesses the initial transition to the *intermediate goal* $g_w$.[2] The function $r(g_w, s_w)$ denotes the reward obtained for reaching the intermediate goal $g_w$ from the intermediate state $s_w$.

To optimize the evidence lower bound, a prevalent approach is the application of the Expectation-Maximization (EM) algorithm (Blei et al., 2017; Zhang et al., 2020). Within the context of our framework, the E-step involves determining the optimal $q^\star$, representing the best approximation to the posterior distribution of the waypoints. Subsequently, the M-step focuses on maximizing the expected log-likelihood, considering the probability of both reaching the final goal from the waypoint and attaining the waypoint from the initial state, with the waypoints being sampled from $q^\star(\cdot)$. An analytical solution can be derived for the optimal waypoint distribution:

**Proposition 2.2.** *The optimal waypoint distribution satisfies the following condition:*

$$q^\star(g_w, s_w \mid g, s) = \frac{p^{\pi(\cdot|\cdot,g)}(g \mid s_w) p^{\pi(\cdot|\cdot,g_w)}(g_w \mid s)\exp(r(g_w, s_w))}{\iint p^{\pi(\cdot|\cdot,g)}(g \mid s'_w) p^{\pi(\cdot|\cdot,g'_w)}(g'_w \mid s)\exp(r(g'_w, s'_w))\mathrm{d}g'_w \mathrm{d}s'_w} \tag{2}$$

We provide the proof in Appendix A.2. Intuitively, the best waypoint should ideally be one that is not only reachable from the starting point but also significantly aids in ultimately reaching the final goal, maximizing the overall likelihood of successfully solving the problem.

Nonetheless, the presence of an intractable normalizing constant precludes direct sampling from $q^\star$. Employing (normalized) importance sampling is also intricate in this scenario: (1) deriving a precise estimation for $p^{\pi(\cdot|\cdot,g)}(g|s)$ is intricate, and (2) developing a proposal distribution that retains information comparable to $q^\star$ is not straightforward. This demands a meticulous assessment of the suitability of a subgoal, considering both the existing state of the policy model and the probability of reaching the final goal.

### 2.3 SEQUENTIAL SUBGOAL OPTIMIZATION

We propose SEGO to tackle the aforementioned challenges, drawing inspiration from Annealed Importance Sampling (Neal, 2001). The innovation of SEGO primarily lies in the E-step, where it is strategically designed to efficiently navigate through vast problem-solving spaces, ensuring it does not get trapped in suboptimal subgoals. For the sake of clarity, we introduce some notations that will be consistently used throughout this section. Specifically, we define $\omega$ as the tuple $(g_w, s_w)$ and use $q^\star(\omega)$ as shorthand for $q^\star(g_w, s_w \mid g, s_0)$.

Beyond the reward model, three additional auxiliary modules are introduced in SEGO to aid in the search for subgoals and to estimate the likelihood $p^{\pi(\cdot|\cdot,g)}(g|s)$. These include the subgoal generator $f(\cdot|g, s)$, the subgoal optimizer $h(\cdot|\omega_{j+1}, g, s)$, and the likelihood estimator $M(g, s)$. The subgoal generator takes the goal and state and produces an initial "draft" subgoal, $\omega$. The process of subgoal sampling progresses in an iterative and sequential manner. In each iteration, the subgoal optimizer proposes a potentially improved subgoal, $\tilde{\omega}$. The improvement is then assessed by considering feedback from the other modules. Intuitively, a subgoal that can attain a higher likelihood of success is regarded as improved. If $\tilde{\omega}$ is indeed an improvement, it is accepted as $\omega$; otherwise, the iteration proceeds with the existing $\omega$.

To rigorously describe this process, we first define a series of functions and transition operators:

---

[2]In this work, a *waypoint* or *subgoal* is defined as the pair $(g_w, s_w)$. These terms are used indiscriminately to refer to intermediate steps aiding in navigation through the state space toward the final goal.

**Definition 1.** *We introduce $f_j(\cdot)$ for $j \in \{0, 1, \ldots, \eta-1\}$ as a weighted blend of $f_0(\cdot)$ and $f(\cdot|g, s)$, given by $f_j(\omega) = f_0(\omega)^{\beta_j} f(\omega|g, s)^{1-\beta_j}$. The sequence of weights $\beta_j$ satisfies $1 = \beta_0 > \beta_1 > \ldots > \beta_\eta = 0$. Specifically, $f_0(\omega)$ satisfies $\frac{f_0(\omega)}{Z_f} = q^\star(\omega)$ where $Z_f$ is the normalizing constant.*

**Definition 2.** *Let $T_j(\omega, \omega')$ for $j \in \{1, \ldots, \eta - 1\}$ denote a transition operator, formulated as $T_j(\omega, \omega') = h(\omega'|\omega) \min\left(1, \frac{f_j(\omega')h(\omega|\omega')}{f_j(\omega)h(\omega'|\omega)}\right)$.*

Then the process of sequentially sampling subgoals is defined as follows:

**Definition 3.** *Let the process start with the sampling of $\omega_{\eta-1}$ from $f(\cdot|g, s)$. Sequentially, $\omega_{\eta-2}$ is derived from $\omega_{\eta-1}$ via the transition operator $T_{\eta-1}$, perpetuating this mechanism until $\omega_0$ is obtained from $\omega_1$ through $T_1$. The joint distribution probability is articulated as $\frac{g(\omega_0, \ldots, \omega_{\eta-1})}{Z_g}$, wherein $g(\omega_0, \ldots, \omega_{\eta-1}) = f(\omega_{\eta-1}|g, s)T_{\eta-1}(\omega_{\eta-1}, \omega_{\eta-2}) \ldots T_1(\omega_1, \omega_0)$ and $Z_g$ is the normalization constant.*

Intuitively, the transition operator, $T_j(\omega, \omega')$, steers the process of picking subgoals and ensuring they are on the right track. It steps in, especially when the agent seems to be heading towards less helpful subgoals. The term $\frac{f_j(\omega')}{f_j(\omega)}$ offers valuable insights into whether a subgoal is beneficial, serving as a helpful gauge for the chosen path's compatibility with the final goal. Conversely, the term $\frac{h(\omega|\omega')}{h(\omega'|\omega)}$ serves as a corrective mechanism, allowing the agent to rethink and correct its path, avoiding situations where it might get stuck in seemingly promising but ultimately unhelpful states.

We further present a proposition that delineates the computation of importance weights attributed to each final subgoal in the sequential process and derivation of an unbiased estimator for the normalization constant $Z_f$:

**Proposition 2.3.** *Consider a collection of sequences, each consisting of $\omega_0, \omega_1, \ldots, \omega_{\eta-1}$. Let $N$ represent the total number of such sequences in the collection. The weight $\alpha$ for each sequence is given by $\alpha = \prod_{j=1}^{\eta} \frac{f_{j-1}(w_{j-1})}{f_j(\omega_{j-1})}$. The unbiased estimator $\hat{Z}_f$ for $Z_f$ is given by:*

$$\hat{Z}_f = \frac{1}{N} \sum \alpha \tag{3}$$

We provide the full proof of the unbiasedness in Appendix A.3. The likelihood estimator $M(g, s)$ is then trained to predict the estimated value $\hat{Z}_f$. SEGO effectively addresses the intricacies of normalized importance sampling, overcoming challenges in estimating $p^{\pi(\cdot|\cdot, g)}(g|s)$ and in formulating informative proposal distributions comparable to $q^\star$. It employs sequential subgoal sampling and sophisticated transition criteria to avoid suboptimal paths and ensure the selection of achievable subgoals. Through meticulous evaluation of each subgoal, considering its alignment with the prevailing state of the policy model and its contribution to the likelihood of achieving the final goal, SEGO optimizes the traversal through complex problem-solving spaces.

Details regarding the implementation and learning objectives of each module can be found in Appendix B, and the comprehensive training algorithm can be found in Appendix C.

## 3 EXPERIMENTS

### 3.1 DATASET AND EVALUATION

**Evaluation Dataset.** To evaluate our proposed model, we employ the GSM8K (Cobbe et al., 2021) and MATH (Hendrycks et al., 2021) datasets. GSM8K is made up of $8,792$ samples, with $1,319$ allocated for testing. It is specifically oriented towards math word problems for elementary school students. Conversely, the MATH dataset assembles a collection of advanced mathematical problems, covering a total of $12,500$ problems, of which $5,000$ are designated for testing. The problems in MATH mainly originate from prestigious competitions such as the American Mathematics Competitions (AMC) and the American Invitational Mathematics Examination (AIME), enabling a thorough evaluation of the model's symbolic reasoning and analytical problem-solving capabilities. For the preprocessing of data in both datasets, we adhere to the methodologies described in their respective original works, ensuring consistency and reliability in our evaluation framework.

**Training Dataset.** The construction of the training dataset for SEGO utilizes the training sets of GSM8K (Cobbe et al., 2021), MATH (Hendrycks et al., 2021), and AQuA (Ling et al., 2017). To initialize the policy model, solutions for each problem are generated using `gpt-3.5-turbo-0613-4k`. We retain only the samples that yield at least one correct answer, resulting in $10,374$ samples for GSM8K, $10,981$ for MATH, and $35,355$ for AQuA. These selectively curated problems serve as the foundational basis for training the various modules integrated into our model. More details regarding the training of the modules can be found in Appendix B.

**Evaluation Metric.** We evaluate by comparing the results of the solution generated by the policy model in SEGO to the provided correct answers within the datasets. For evaluation, we report the accuracy, specifically focusing on the rate at which the policy model correctly solves the problems on the first attempt.

## 3.2 BASELINES

**Closed-Source Models.** (1) **GPT-4**: A model that sets a standard in various academic domains, including those that require intricate mathematical reasoning (OpenAI, 2023). (2) **PaLM-2**: A model that excels at logical reasoning and multilingual tasks, demonstrating advanced capabilities in reasoning and solving complex mathematical problems in multiple languages (Anil et al., 2023). (3) **Minerva**: A model that specializes in quantitative reasoning, providing precise and comprehensive solutions to advanced mathematical, scientific, and engineering problems (Lewkowycz et al., 2022).

**Open-Source Models.** (1) **LLaMA2**: A model that is trained on 2 trillion tokens of publicly accessible data, exhibits outstanding capabilities in mathematical reasoning (Touvron et al., 2023). (2) **WizardMATH**: A model that enhances the mathematical reasoning capabilities of LLaMA2 by curating more complex and diverse SFT data (Luo et al., 2023). (3) **CodeLLaMA**: A model that excels in code-related tasks with implications in mathematical programming and algorithm synthesis, demonstrating superior infilling capabilities and support for extensive input contexts in programming tasks (Rozière et al., 2023).[3]

## 3.3 IMPLEMENTATION DETAILS

We maintain model consistency by employing CodeLLaMA as the base model for both the policy model and auxiliary modules, including the Subgoal Generator, Subgoal Optimizer, Reward Model, and Likelihood Estimator. Efficient finetuning of the auxiliary modules is achieved through the utilization of LoRA (Hu et al., 2021), configured with parameters $r = 16$, lora_alpha $= 32$, and lora_dropout $= 0.05$, targeting the "q_proj" and "k_proj" modules. The learning rates are set at $1e - 5$ and $1e - 4$ for the policy and auxiliary modules, respectively, with a uniform batch size of $32$. When collecting data from `gpt-3.5-turbo-0613`, we set temperature and top_p as $0.8$ and $1.0$ respectively. All models go through an initial training phase of $4,800$ steps. Subsequently, a sequential optimization process is conducted, with the number (N) and length ($\eta$) of sequences set as 2 and 3 respectively, and the temperature and top_p for the Subgoal GeneratorOptimizer and the policy model configured at $0.2$ and $0.95$ respectively. This optimization is performed three times, each lasting $1,200$ steps, and when $\eta = 3$, the parameters $\beta_1$ and $\beta_2$ are precisely set at $0.33$ and $0.66$ respectively. Rigorous contamination checking, as delineated by OpenAI (2023), is executed to verify the purity of our test sets for GSM8K and MATH. During the test phase, a greedy search strategy is employed.

## 3.4 MAIN RESULTS

We follow Drori et al. (2022) and Chen et al. (2022) to employ the program of though (PoT) to solve math problems. The experiment results, as shown in Table 1, yield several key observations. First, SEGO demonstrates remarkable capability in solving problems from the GSM8K and MATH datasets. Specifically, SEGO (7B) attains accuracies of 68.7% and 36.8% on GSM8K and MATH, respectively, while SEGO (13B) achieves accuracies of 72.5% and 40.0% on the same datasets.

---

[3]For CodeLLaMA, we ensure consistency with our models by employing identical decoding methods and prompts during implementation, while for the other models, we refer to the results reported in their respective papers.

Table 1: Evaluation results on GSM8K and MATH.

| Model | Base | Prompt | Params | GSM8K | MATH |
|---|---|---|---|---|---|
| GPT-4 | - | CoT | - | 92.0 | 42.5 |
| PaLM-2 | PaLM | CoT | 540B | 80.7 | 34.3 |
| Minerva | PaLM | CoT | 540B | 58.8 | 33.6 |
| LLaMA2 | LLaMA2 | CoT | 7B | 14.6 | 2.5 |
| | | | 13B | 28.7 | 3.9 |
| WizardMATH | LLaMA2 | CoT | 7B | 54.9 | 10.7 |
| | | | 13B | 63.9 | 14.0 |
| CodeLLaMA | CodeLLaMA | PoT | 7B | 25.2 | 14.2 |
| | | | 13B | 36.1 | 18.1 |
| **SEGO (ours)** | **CodeLLaMA** | **PoT** | **7B** | **68.7** | **36.8** |
| | | | **13B** | **72.5** | **40.0** |

These results are superior to those of all other models of comparable sizes, highlighting the efficacy of SEGO in mathematical problem-solving tasks.

Second, supervised finetuning and the program of thought (PoT) approach contribute to enhanced performance in the tested models. Models benefiting from supervised finetuning, such as Wizard-MATH and SEGO, exhibit improved performance by adapting to the specific characteristics and requirements of the task. Meanwhile, the PoT approach excels in tasks of higher complexity, particularly evident in the MATH dataset, as seen in the comparison between SEGO and WizardMATH, and CodeLLaMA and LLaMA2. The structured stepwise reasoning facilitated by programs enables models to effectively leverage Python tools, augmenting their capability to solve intricate problems.

Lastly, the integration of Sequential Subgoal Optimization plays an important role in elevating the performance of SEGO. This enhancement underscores the importance of strategic subgoal optimization in tackling mathematical problems, allowing a more effective navigation to reach accurate solutions.

Table 2: Ablation Study Results on GSM8K and MATH datasets.

| Models | GSM8K | MATH |
|---|---|---|
| Ours | 68.7 | 36.8 |
| -Sequential | 61.3 | 34.9 |
| -Sequential & Subgoal | 57.1 | 32.6 |
| -Sequential & Subgoal & SFT | 25.2 | 14.2 |

## 4 ANALYSIS

### 4.1 ABLATION STUDY

To gain insights into the contributions of each component within our framework, we conduct ablation experiments on 7B CodeLLaMA with SEGO and three ablated versions. The ablated versions are as follows: (1) **-Sequential:** By omitting the sequential subgoal optimization, this version highlights the impact of sequential optimization on the model's proficiency. (2) **-Sequential & Subgoal:** This version, devoid of the subgoal and relying solely on supervised finetuning, sheds light on the unique contribution of the subgoal to problem-solving efficacy. (3) **-Sequential & Subgoal & SFT:** As the most reduced version, absent of both the subgoal and supervised finetuning, it essentially mirrors the base 7B CodeLLaMA.

The ablation results in Table 2 reveal the critical impact of sequential subgoal optimization in SEGO. The absence of this component in the **-Sequential** variant results in a discernible reduction in ac-

curacy, highlighting its crucial role in bolstering the model's ability to solve complex problems effectively. Additionally, the substantial decrement in performance observed in the **-Sequential & Subgoal & SFT** variant, equivalent to the base 7B CodeLLaMA, demonstrates the collective contribution of sequential subgoal optimization along with other components in elevating SEGO's problem-solving prowess in mathematical domains.

## 4.2 ANALYSIS OF HYPERPARAMETERS

In this section, we conduct a detailed examination of the hyperparameters $N$ and $\eta$, where $N$ represents the number of sequences and $\eta$ denotes the length of each sequence, as defined in Proposition 2.3. All the experiments in this section are anchored on the 7B CodeLLaMA to ensure consistency in the results.

**The balance between $N$ and $\eta$.** We begin by exploring various combinations of $N$ and $\eta$, illustrated in Figure 2, to comprehend the synergistic effects of these parameters on the model's performance. The results on GSM8K and MATH reveal that incrementing both $N$ and $\eta$ typically enhances the model's accuracy, achieving $68.7\%$ on GSM8K and $36.8\%$ on MATH at $N = 2$ and $\eta = 3$. However, the enhancements appear to stabilize beyond certain thresholds, indicating optimal points for these parameters.

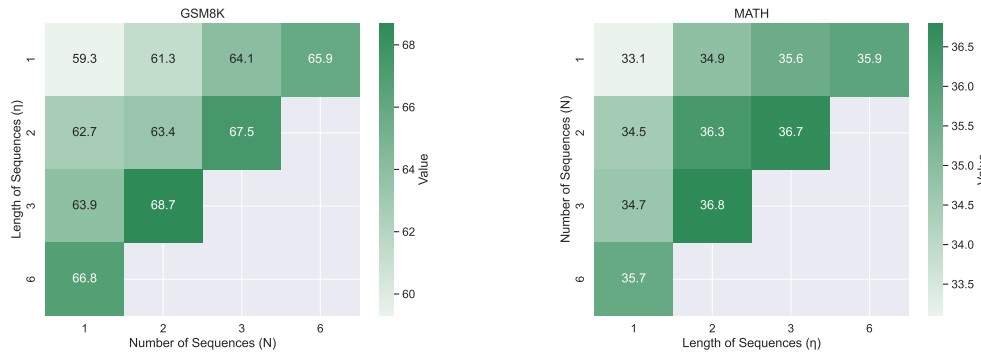

Figure 2: The balance between the number of sequences ($N$) and the length of sequences ($\eta$) on the test sets of GSM8K and MATH.

**In-depth analysis of Hyperparameters $N$ and $\eta$.** We further conduct an in-depth analysis of the hyperparameters $N$ and $\eta$, examining each one's individual impact by holding one constant and varying the other. The results are illustrated in Figure 3. From the results, it is clear that when $N = 2$, the model achieves peak accuracy at $\eta = 3$ for both GSM8K and MATH, with no significant gains beyond this point. Similarly, with $\eta = 3$, optimal accuracy is reached at $N = 2$, remaining stable thereafter.

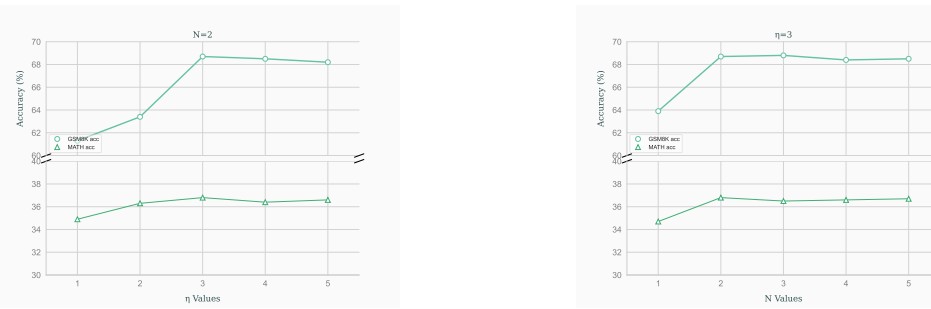

Figure 3: Analysis of model accuracy for variations $N$ and $\eta$. Left: Fixed $N = 2$ and various $\eta$; Right: Fixed $\eta = 3$ and various $N$.

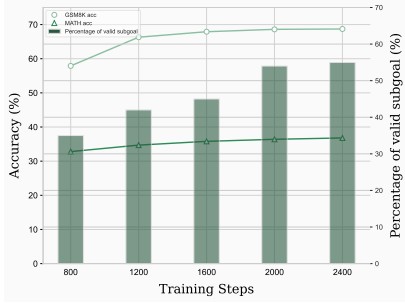 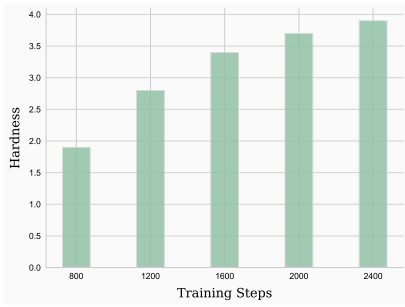

Figure 4: Left: Changes in the percentage of valid subgoals during the RL training. Right: Changes in hardness of problems yielding valid subgoals.

### 4.3 ANALYSIS OF SUBGOAL EVOLUTION

**Validity and Progression of Subgoals.**    To deepen our understanding of subgoals during the Reinforcement Learning phase, we analyze the evolution of subgoal validity and its correlation with the performance on the test set. A subgoal (i.e., $g_w$ and $s_w$) is deemed valid if both $\tau_1$ and $\tau_2$, sampled with policies $\pi(\cdot|s_w, g)$ and $\pi(\cdot|s, g_w)$, yield correct solutions for goals $g$ and $g_w$ respectively. Our findings, illustrated in Figure 4 (Left), reveal a positive correlation between the progression of training steps and the percentage of valid subgoals. This increase in valid subgoals is paralleled by improvements in accuracy on both GSM8K and MATH datasets, suggesting that the validity of subgoals is a crucial factor in enhancing the model's problem-solving capabilities.

**Hardness of Problems Yielding Valid Subgoals.**    To further our exploration of subgoals, we delve into the relationship between the hardness of problems and the emergence of valid subgoals. This analysis aims to reveal any trends in the difficulty of problems that tend to yield valid subgoals, providing nuanced insights into the learning progression. The hardness of each problem is labeled by ChatGPT, with more details available in Appendix D. Our findings, shown in Figure 4 (Right), reveal a correlation between training progression and the model's ability to formulate valid subgoals for increasingly intricate problems, underscoring its evolving sophistication and adaptability in problem-solving.

## 5 RELATED WORKS

### 5.1 MATHEMATICAL REASONING IN LARGE LANGUAGE MODELS

The exploration of mathematical reasoning in Large Language Models (LLMs) has been significantly influenced by the development of datasets such as GSM8K (Cobbe et al., 2021) and MATH (Hendrycks et al., 2021), serving as crucial benchmarks for assessing machine learning models in mathematical domains. GSM8K encompasses a variety of grade school math problems, while MATH compiles challenging competition mathematics problems. The introduction of extensive datasets (Koncel-Kedziorski et al., 2016; Ling et al., 2017; Talmor et al., 2018; Geva et al., 2021) and platforms like MWPToolkit (Lan et al., 2022) has enriched the field. This exploration is systematically categorized into two main domains: prompting strategies and learning with verifications. In the realm of prompting strategies, a variety of methods have been conceptualized to enhance the reasoning capabilities of LLMs. Techniques such as Chain-of-Thought Prompting (Wei et al., 2023; Wang et al., 2022), Progressive-Hint Prompting (Zheng et al., 2023), and Least-to-Most Prompting (Zhou et al., 2022) have been instrumental in progressively guiding LLMs to accurate conclusions and facilitating the generation of intermediate reasoning steps. Moreover, methodologies like Complexity-Based Prompting (Fu et al., 2023) and Self-Consistency(Wang et al., 2022) exploit higher reasoning complexity and diverse reasoning paths, respectively, to realize significant advancements in multi-step reasoning tasks. Within learning with verifications, the emphasis is on optimizing the mathematical proficiencies of LLMs through the integration of verifiers. Strategies like outcome-based verifiers (Cobbe et al., 2021), step-aware verifiers (Li et al., 2023; Lightman et al., 2023), and learning from partially-correct solutions (Ni et al., 2023) have been deployed to

bolster reliability and precision in mathematical reasoning. While the aforementioned domains have significantly advanced mathematical reasoning within LLMs, our approach is orthogonal to these categories. We concentrate on the formulation of adaptive curricula, emphasizing the incorporation of subgoals, to facilitate nuanced learning pathways and enhance the model's mathematical reasoning capabilities. A parallel and notably concurrent work, MAmmoTH (Yue et al., 2023), investigates the impact of instruction finetuning to empower large language models with mathematical problem-solving capabilities. This can be considered as an implementation of the instruction finetuning stage within our framework, with further discussions and experimental results provided in Appendix E.

## 5.2 SUBGOAL SEARCH IN REINFORCEMENT LEARNING

Subgoal Search is a central component in reinforcement learning, essential for empowering AI systems to navigate through complex, extensive tasks effectively. This concept has played a vital role in uncovering important aspects such as the benefits of recognizing and rewarding subgoals (Zhai et al., 2022), the proper structuring of Markov decision processes for hierarchical reinforcement learning (Wen et al., 2020), the difficulties in selecting the most suitable options for planning (Jinnai et al., 2019a), and the incorporation of temporal abstraction in RL (Fruit et al., 2017). The practical research in this field mainly focuses on exploring and creating subgoals for planning and developing learning curricula for subgoals. Exploration is aimed at finding the best or most efficient strategies, using diverse approaches like reducing cover time (Jinnai et al., 2019b), understanding dynamical distances (Hartikainen et al., 2019), increasing entropy (Pitis et al., 2020), and applying asymmetric self-play (OpenAI et al., 2021). In the area of subgoal planning, a variety of algorithms have been developed to refine decision-making processes. For example, SoRB (Eysenbach et al., 2019) utilizes RL to develop a graph for subgoal sequences, DC-MCTS (Parascandolo et al., 2020) employs learned subgoal proposals to divide tasks, PAIR (Li et al., 2022) combines online RL with offline supervised learning, and (Moro et al., 2022) improve MCTS with Hindsight Experience Replay for goal-oriented planning. Moreover, the work by (Chane-Sane et al., 2021) provides concise insights into improving goal-conditioned reinforcement learning by conceptualizing imagined subgoals, adding a fresh viewpoint to the field. Research in curriculum learning has developed innovative methods to construct curricula that systematically escalate the complexity of subgoals, thereby improving the speed and quality of learning (Zhang et al., 2020; 2021). The exploration of subgoal learning in the realm of complex mathematical problem-solving represents a largely unexplored field. Our work delves into the inherent challenges of applying subgoal learning in mathematical contexts, specifically, the difficulty in identifying the optimal subgoal within expansive state spaces, and introduces a theoretical framework to navigate these challenges.

## 6 CONCLUSION

In conclusion, we have introduced SEGO, a novel framework for enhancing the problem-solving capabilities of Large Language Models (LLMs) in the context of mathematical tasks. Drawing inspiration from the success of subgoal-based methods, SEGO establishes a theoretical connection between the subgoal breakdown process and the probability of solving problems. By generating problem-specific subgoals and adjusting them according to carefully designed criteria, our framework significantly improves the performance of LLMs in solving mathematical problems. Through experiments conducted on two primary benchmarks, GSM8K and MATH, we have demonstrated the efficacy of SEGO in outperforming existing methods with comparable model sizes. These results not only showcase the potential of our proposed framework in advancing the field of AI-driven mathematical problem-solving but also highlight the importance of strategic subgoal optimization in tackling complex problems.

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

# A PROOFS

## A.1 PROOF OF PROPOSITION 2.1

In this subsection, we establish the proof of Proposition 2.1

*Proof.* We start by considering the joint distribution $p(g, s_w, g_w|s)$, which can be factorized as $p^{\pi(\cdot|\cdot,g)}(g \mid s_w)p^{\pi(\cdot|\cdot,g_w)}(g_w|s)p(s_w|g_w)$.

The log-likelihood of reaching the goal $g$ from $s$ can be expressed as:

$$\log p^{\pi(\cdot|\cdot,g)}(g|s) = \log \mathbb{E}_{q(g_w,s_w|g,s)}\left[\frac{p(g, s_w, g_w|s)}{q(g_w, s_w|g, s)}\right] \tag{4}$$

Expanding the expectation, we get:

$$\log p^{\pi(\cdot|\cdot,g)}(g|s) = \log \iint q(g_w, s_w|g, s)\frac{p(g, s_w, g_w|s)}{q(g_w, s_w|g, s)}\mathrm{d}g_w\mathrm{d}s_w \tag{5}$$

Utilizing Jensen's inequality, we establish a lower bound for the log-likelihood as follows:

$$\log p^{\pi(\cdot|\cdot,g)}(g|s) \geq \mathbb{E}q(g_w, s_w|g, s)\left[\log p^{\pi(\cdot|\cdot,g)}(g|s_w) + \log p^{\pi(\cdot|\cdot,g_w)}(g_w|s) \right. \\ \left. + \log p(s_w|g_w) - \log q(g_w, s_w|g, s)\right] \tag{6}$$

Given that $\log p(s_w|g_w) = r(g_w, s_w) - \log\left(\sum_{s'_w}\exp(r(g_w, s'_w))\right)$, we introduce $C$ such that:

$$C = \log\left(\sum_{s'_w}\exp(r(g_w, s'_w))\right) \tag{7}$$

Since $C$ is a constant, it can be absorbed into the lower bound $\mathcal{L}$ as a constant term, which does not affect the optimization process. Therefore, the lower bound $\mathcal{L}$ can be written as:

$$\mathcal{L} = \mathbb{E}_{q(g_w,s_w|g,s)}\left[\log p^{\pi(\cdot|\cdot,g)}(g|s_w) + \log p^{\pi(\cdot|\cdot,g_w)}(g_w|s) + r(g_w, s_w) - \log q(g_w, s_w|g, s)\right] \tag{8}$$

This completes the proof of proposition 2.1. □

## A.2 PROOF OF PROPOSITION 2.2

In this subsection, we establish the proof of Proposition 2.2

*Proof.* The optimization objective for finding $q(g_w, s_w \mid g, s)$ is:

$$\mathbb{E}_{q(g_w,s_w|g,s)}\left[\log p^{\pi(\cdot|\cdot,g)}(g \mid s_w) + \log p^{\pi(\cdot|\cdot,g_w)}(g_w \mid s) + r(g_w, s_w) - \log q(g_w, s_w \mid g, s)\right] \tag{9}$$

Introducing a Lagrange multiplier $\lambda$, the Lagrangian $\mathcal{J}$ is constructed as:

$$\mathcal{J} = \mathbb{E}q(g_w, s_w \mid g, s)\left[\log p^{\pi(\cdot|\cdot,g)}(g \mid s_w) + \log p^{\pi(\cdot|\cdot,g_w)}(g_w \mid s) + r(g_w, s_w) \right. \\ \left. - \log q(g_w, s_w \mid g, s)\right] + \lambda\left(\int q(g_w, s_w \mid g, s)\mathrm{d}g_w\mathrm{d}s_w - 1\right) \tag{10}$$

Differentiating $\mathcal{J}$ with respect to $q(g_w, s_w \mid g, s)$ and setting it to zero yields:

$$\log p^{\pi(\cdot|\cdot,g)}(g \mid s_w) + \log p^{\pi(\cdot|\cdot,g_w)}(g_w \mid s) + r(g_w, s_w) - \log q(g_w, s_w \mid g, s) - 1 + \lambda = 0 \quad (11)$$

Simplifying, we get:

$$q(g_w, s_w \mid g, s) = \exp(\lambda - 1)p^{\pi(\cdot|\cdot,g)}(g \mid s_w)p^{\pi(\cdot|\cdot,g_w)}(g_w \mid s)\exp(r(g_w, s_w)) \quad (12)$$

To ensure $q(g_w, s_w \mid g, s)$ is a valid probability distribution, it is normalized as:

$$q^\star(g_w, s_w \mid g, s) = \frac{p^{\pi(\cdot|\cdot,g)}(g \mid s_w)p^{\pi(\cdot|\cdot,g_w)}(g_w \mid s)\exp(r(g_w, s_w))}{\iint p^{\pi(\cdot|\cdot,g)}(g \mid s'_w)p^{\pi(\cdot|\cdot,g'_w)}(g'_w \mid s)\exp(r(g'_w, s'_w))\mathrm{d}g'_w\mathrm{d}s'_w} \quad (13)$$

The denominator serves as the normalizing constant, ensuring that $q^\star(g_w, s_w \mid g, s)$ sums to one over its domain, thereby satisfying the properties of a probability distribution.

This concludes the proof. $\qquad \square$

### A.3 PROOF OF PROPOSITION 2.3

To establish the validity of the proposition, we begin by proving three essential lemmas:

**Lemma 1.** *Given $f_j(\omega)$ and $T_j(\omega, \omega')$ as specified in definition 2, it holds that*

$$\int T_j(\omega, \omega') \, \mathrm{d}\omega' = 1. \quad (14)$$

*Proof.* Starting with the definition of $T_j(\omega, \omega')$,

$$T_j(\omega, \omega') = h(\omega'|\omega) \min\left(1, \frac{f_j(\omega')h(\omega|\omega')}{f_j(\omega)h(\omega'|\omega)}\right),$$

we can derive the following inequalities:

Firstly,

$$\begin{aligned}
\int T_j(\omega, \omega') \, \mathrm{d}\omega' &\geq \int h(\omega'|\omega)\frac{f_j(\omega')h(\omega|\omega')}{f_j(\omega)h(\omega'|\omega)} \, \mathrm{d}\omega' \\
&= \int \frac{f_j(\omega')h(\omega|\omega')}{f_j(\omega)} \, \mathrm{d}\omega' \\
&= \frac{Z_{f_j}}{f_j(\omega)} \int p(\omega')h(\omega|\omega') \, \mathrm{d}\omega' \\
&= \frac{Z_{f_j}}{f_j(\omega)} \int p(\omega, \omega') \, \mathrm{d}\omega' \\
&= \frac{Z_{f_j}}{f_j(\omega)}p(\omega) \\
&= 1,
\end{aligned}$$

where $Z_{f_j}$ is the normalizing constant of $f_j(\omega)$.

Secondly,

$$\int T_j(\omega, \omega') \, \mathrm{d}\omega' \leq \int h(\omega'|\omega) \, \mathrm{d}\omega' = 1.$$

Combining these inequalities, we conclude that

$$\int T_j(\omega, \omega') \, \mathrm{d}\omega' = 1.$$

$\qquad \square$

**Lemma 2.** *Let $f_j(\omega)$ and $T_j(\omega, \omega')$ be as specified in Definition 2. Define $p_j(\omega)$ as*

$$p_j(\omega) = \frac{f_j(\omega)}{\int f_j(\omega') \, d\omega'}.$$

*Then, the following detailed balance condition holds:*

$$p_j(\omega) T_j(\omega, \omega') = p_j(\omega') T_j(\omega', \omega). \tag{15}$$

*Proof.* The proof can be divided into two cases:

**Case 1:** $p_j(\omega')h(\omega \mid \omega') > p_j(\omega)h(\omega' \mid \omega)$

Starting with $p_j(\omega')T_j(\omega', \omega)$, we have:

$$
\begin{aligned}
p_j(\omega')T_j(\omega', \omega) &= \cancel{p_j(\omega')h(\omega \mid \omega')}\frac{p_j(\omega)h(\omega' \mid \omega)}{\cancel{p_j(\omega')h(\omega \mid \omega')}} \\
&= p_j(\omega)h(\omega' \mid \omega) \\
&= p_j(\omega)T_j(\omega, \omega').
\end{aligned}
$$

**Case 2:** $p_j(\omega')h(\omega \mid \omega') \le p_j(\omega)h(\omega' \mid \omega)$

Starting with $p_j(\omega)T_j(\omega, \omega')$, we have:

$$
\begin{aligned}
p_j(\omega)T_j(\omega, \omega') &= \cancel{p_j(\omega)h(\omega' \mid \omega)}\frac{p_j(\omega')h(\omega \mid \omega')}{\cancel{p_j(\omega)h(\omega' \mid \omega)}} \\
&= p_j(\omega')h(\omega \mid \omega') \\
&= p_j(\omega')T_j(\omega', \omega).
\end{aligned}
$$

In both cases, we find that $p_j(\omega)T_j(\omega, \omega') = p_j(\omega')T_j(\omega', \omega)$, thereby proving the lemma. □

**Lemma 3.** *Let $f_j(\omega)$ and $T_j(\omega, \omega')$ be as defined in Definition 2. Define the normalized distribution $p_j(\omega)$ as*

$$p_j(\omega) = \frac{f_j(\omega)}{\int f_j(\omega') \, d\omega'}.$$

*Then, $T_j(\omega, \omega')$ preserves the invariance of $p_j(\omega)$, formally defined as*

$$\int T_j(\omega', \omega)p_j(\omega') \, d\omega' = p_j(\omega).$$

*Proof.* We proceed by leveraging the results from Lemma 1 and Lemma 2. Specifically, we have:

$$
\begin{aligned}
\int T_j(\omega', \omega)p_j(\omega') \, d\omega' &= \int T_j(\omega, \omega')p_j(\omega) \, d\omega' \\
&= p_j(\omega) \int T_j(\omega, \omega') \, d\omega' \\
&= p_j(\omega).
\end{aligned}
$$

This confirms that $T_j(\omega, \omega')$ preserves the invariance of $p_j(\omega)$, thereby proving Lemma 3. □

Now we give the proof of Proposition 2.3.

*Proof.* We first define the function $f$ as follows:

$$f(\omega_0, \dots, \omega_{\eta-1}) = \frac{f_0(\omega_0)}{f_1(\omega_0)}T_1(\omega_1, \omega_0) \dots \frac{f_{\eta-2}(\omega_{\eta-2})}{f_{\eta-1}(\omega_{\eta-2})}T_{\eta-1}(\omega_{\eta-1}, \omega_{\eta-2})f_{\eta-1}(\omega_{\eta-1})$$

Given the definition of $Z_f$, we have

$$Z_f = \int f_0(\omega) \, \mathrm{d}\omega$$

By Lemma 3, we have:

$$\int T_j(\omega_j, \omega_{j-1}) f_j(\omega_j) \, \mathrm{d}\omega_j = f_j(\omega_{j-1})$$

Thus, we can write:

$$
\begin{aligned}
&\int \frac{f(\omega_0, \cdots, \omega_{\eta-1})}{Z_f} \mathrm{d}\omega_0 \cdots \mathrm{d}\omega_{\eta-1} \\
&= \int \frac{f_0(\omega_0)}{Z_f} \mathrm{d}\omega_0 \int \frac{T_1(\omega_1, \omega_0) f_1(\omega_1)}{f_1(\omega_0)} \mathrm{d}\omega_1 \cdots \int \frac{T_{\eta-1}(\omega_{\eta-1}, \omega_{\eta-2}) f_{\eta-1}(\omega_{\eta-1})}{f_{\eta-1}(\omega_{\eta-2})} \mathrm{d}\omega_{\eta-1} \\
&= \int \frac{f_0(\omega_0)}{Z_f} \mathrm{d}\omega_0 \\
&= 1
\end{aligned}
$$

This implies that $Z_f$ is also the normalizing constant of $f(\omega_0, \ldots, \omega_{\eta-1})$.

Since $f_n(\cdot)$ is a distribution, it is evident that $Z_g = 1$.

We have:

$$
\begin{aligned}
&\mathbb{E}_{g(\cdot)} \left[ \frac{1}{N} \sum \alpha \right] \\
&= \mathbb{E}_{g(\cdot)} \alpha \\
&= \mathbb{E}_{g(\cdot)} \frac{f(\omega_0, \cdots, \omega_{\eta-1})}{g(\omega_0, \cdots, \omega_{\eta-1})} \\
&= Z_f \int \frac{f(\omega_0, \cdots, \omega_{\eta-1})}{Z_f} \mathrm{d}\omega_0 \cdots \mathrm{d}\omega_{\eta-1} \\
&= Z_f
\end{aligned}
$$

This concludes the proof of Proposition 2.3. $\qquad\square$

## B  MORE IMPLEMENTATION DETAILS FOR EACH MODULE

The framework of SEGO is composed of five pivotal modules, each serving a distinct purpose to enhance the system's overall efficacy. The module $f(\cdot|g, s)$ acts as the Subgoal Generator, formulating intermediate objectives. $h(\cdot|\omega, g, s)$ serves as the Subgoal Optimizer, refining the generated subgoals for optimality. $r(g, w)$ is the Reward Model, assessing the associated rewards for each subgoal. $\pi(\cdot|s, g)$ operates as the Policy Model, determining optimal actions based on the current state and subgoal. Lastly, $M(g, s)$ functions as the likelihood estimator, computing the likelihood of a subgoal given the current state of policy model (i.e., $p^{\pi(\cdot|\cdot,g)}(g|s)$ in Eq.(1)).

### B.1  SUBGOAL GENERATOR

The subgoal generator is trained through instruction finetuning, utilizing data collected from `gpt-3.5-turbo-0613`. The instruction template is defined as:

```
Break down the given problem into a smaller task (a subproblem)
and devise a method to solve it, considering a provided partial
solution to the original problem as a starting point.

### Input:
```

```
{problem}

{partial solution}

### Output:
{subproblem}{solution}[EOS]
```

This module, fundamentally built on the architecture of CodeLLaMA Rozière et al. (2023), leverages the capabilities of LoRA Hu et al. (2021) for efficient finetuning. The primary objective is to accurately predict {subproblem}{solution}[EOS] from its preceding context, realized through a causal language modeling.

## B.2 SUBGOAL OPTIMIZER

The subgoal optimizer is also trained through instruction finetuning, drawing upon data from gpt-3.5-turbo-0613. The instruction template for this module is as follows:

```
Optimize the given subproblem to make it more manageable. Then,
develop a method to solve it, considering a provided partial solution
to the original problem as a starting point.

### Input:
{problem}

{partial solution}

{subproblem}{solution}

### Output:
{optimized subproblem}{optimized solution}[EOS]
```

This module, also built on CodeLLaMA, utilizes LoRA for efficient parameter finetuning. The aim here is to accurately predict {optimized subproblem}{optimized solution}[EOS] from the provided context, ensuring the outputs are coherent and contextually aligned.

## B.3 REWARD MODEL

The reward model $r(g, s)$ serves as a discriminative model, designed to determine the probability of $s$ being a correct solution to $g$. This model is built on the architecture of CodeLLaMA and employs LoRA to achieve efficient finetuning. To train this model, it is imperative to construct both positive examples, where $s$ is a correct solution to $g$, and negative examples, where $s$ is not a correct solution to $g$. This is achieved by utilizing the policy model, post its warm-up phase, to generate solutions for each corresponding problem. Given that each problem is accompanied by a human-annotated answer, solutions leading to the correct answer are categorized as positive examples, while those not leading to the correct answer are treated as negative examples. The reward model is trained through instruction finetuning, utilizing the following instruction template:

```
Does the provided solution accurately address the given problem?
{problem} {solution} {Y/N}.
```

For positive and negative examples, the model acts as a conditional language model, predicting Y for positive and N for negative examples, based on the preceding context.

## B.4 POLICY MODEL

The policy model, denoted as $\pi(\cdot|s_t, g)$, is initially warmed up to emulate the behavior of a sophisticated model, specifically gpt-3.5-turbo-0613 in our scenario. This involves the collection of successful trajectories $(a_0, a_1, \ldots)$ from gpt-3.5-turbo-0613, corresponding to a specific

goal. Here, the trajectory represents a potential solution to the goal $g$. In cases where a human-annotated answer for $g$ is available, only those potential solutions leading to the correct answer are retained. In the absence of such annotated answers, solutions are retained based on the predictions of the reward model. The training of the policy model is conducted through instruction finetuning, utilizing the following instruction template:

```
Construct a Python script to address the given problem:
{problem}

### Response:
{solution}
```

In this template, `problem` and `solution` represent the goal $g$ and the trajectory respectively. The base model for this process is CodeLLaMA, and it undergoes full parameter finetuning to optimize its performance. As the sequential subgoal optimization process progresses, the model is further trained by utilizing self-generated successful trajectories.

### B.5 LIKELIHOOD ESTIMATOR

The likelihood estimator, $M(g, s)$, is designed to approximate the probability $p^{\pi(\cdot|\cdot,g)}(g|s)$, representing the likelihood of reaching goal $g$ from state $s$ while following the strategy of the policy model, $\pi(\cdot|\cdot, g)$. Initially, before the sequential subgoal optimization process, this estimator is trained to approximate the probability of achieving goal $g$ given a state $s_t$, where $s_t$ is the prefix of a successful trajectory corresponding to goal $g$. Such trajectories are sampled from the policy model after it has been warmed up. As the optimization process progresses, the estimator is further trained to approximate the estimated $\hat{Z}_f$, utilizing instruction finetuning. The instruction template is defined as:

```
Determine the probability of resolving the problem, starting from
the partial solution: {problem} {partial solution}.
```

This model, built on the CodeLLaMA architecture, is finetuned using LoRA. It is noted that, during each iteration of the sequential subgoal optimization process, a unique set of LoRA parameters is used to avoid any potential discrepancies between iterations. This approach ensures that the likelihood estimator accurately reflects the real-time capabilities of the policy model.

## C ALGORITHM OVERVIEW

This section provides an overview of the Sequential Subgoal Optimization process, detailed in Algorithm 1. Our method starts with the initialization of various modules, including the Subgoal Generator $f$, Subgoal Optimizer $h$, Reward Model $r$, Policy Model $\pi$, and likelihood estimator $M$, using instruction finetuning to enhance their adaptability to specific task requirements (see Appendix B for details). Following initialization, the Sequential Subgoal Optimization process leverages the interaction of the prepared components to optimize subgoals systematically.

---

**Algorithm 1** Sequential Subgoal Optimization

---

**Requires:**    $f$:    Subgoal Generator
             $h$:    Subgoal Optimizer
             $r$:    Reward Model
             $\pi$:    Policy Model
             $M$:    Likelihood Estimator
             $N_{\max}$:    the maximum number of iterations in an optimization process
             $D_{\text{init}}$:    the dataset for the wamup of policy model

$\mathcal{D}_1, \mathcal{D}_2 \leftarrow \emptyset$
$t \leftarrow 0$
**while** $t < N_{\max}$ **do**
     sample $(g, s)$ from $D_{\text{init}}$
     $\bar{\alpha} \leftarrow 0$
     **for** $i \in 1, \ldots, N$ **do**
         $\omega_\eta^{(i)} \sim f(\cdot | g, s)$
         $\alpha^{(i)} \leftarrow 1$
         **for** $j \in \eta - 1, \ldots, 0$ **do**
             $\omega_j^{(i)} \sim q(\cdot \mid \omega_{j+1}^{(i)}, g, s)$
             Calculate $f_j(\omega_j^{(i)})$ and $f_j(\omega_{j+1}^{(i)})$ following Definition 1
             **if** $\log f_j(\omega_j^{(i)}) + \log h(\omega_{j+1}^{(i)} \mid \omega_j^{(i)}, g, s) < \log f_j(\omega_{j+1}^{(i)}) + \log h(\omega_j^{(i)}, g, s) | \omega_{j+1}^{(i)}$ **then**
                 $\omega_j^{(i)} \leftarrow \omega_{j+1}^{(i)}$
                 $\log f_j(\omega_j^{(i)}) \leftarrow \log f_j(\omega_{j+1}^{(i)})$
             Calculate $f_{j+1}(\omega_j^{(i)})$ following Definition 1
             $\log \alpha^{(i)} \leftarrow \log \alpha^{(i)} + \log f_j(\omega_j^{(i)}) - \log f_{j+1}(\omega_j^{(i)})$
         $\bar{\alpha} \leftarrow \bar{\alpha} + \frac{1}{N} \alpha^{(i)}$
     $\omega_0^{(i)} \sim \text{SoftMax}(\log \alpha^{(i)})$
     Let $\omega_0^{(i)} = (g_{w,0}, s_{w,0})$
     Sample $\tau_1, \tau_2$ with policies $\pi(\cdot | s_{w_0}, g)$ and $\pi(\cdot | s, g_{w,0})$ respectively
     $\mathcal{D}_1 \leftarrow \mathcal{D}_1 \cup \{\tau_1, \tau_2\}, \mathcal{D}_2 \leftarrow \mathcal{D}_2 \cup \{\bar{\alpha}\}$
     $t \leftarrow t + 1$
Train policy model $\pi$ on $\mathcal{D}_1$ and likelihood estimator $M$ on $\mathcal{D}_2$

---

## D THE ANNOTATION OF PROBLEM HARDNESS

We employ the following prompt to automatically annotate the difficulty with `gpt-3.5-turbo-0613`:

```
Please assign a score between 1 and 5 to the following question,
indicating its level of difficulty and complexity. A higher score
should be given to denote greater difficulty and complexity.

Please provide only the score, without any additional explanations
or reasons.

### Input:
{question}

### Output:
```

## E COMPARATION WITH MAMMOTH

Table 3: Comparison with MAmmoTH on GSM8K and MATH.

| Model | Params | GSM8K | MATH |
|-------|--------|-------|------|
| MAmmoTH-Coder | 7B | 58.8 | 35.2 |
| | 13B | 64.3 | 38.6 |
| SEGO (-Sequential & Subgoal) | 7B | 57.1 | 35.9 |
| | 13B | 62.0 | 37.5 |
| **SEGO** | **7B** | **68.7** | **40.9** |
| | **13B** | **72.5** | **44.2** |

In this section, we draw a comparison between our proposed model and MAmmoTH-Coder (Yue et al., 2023), a highly concurrent work emerging post the writing of our manuscript. MAmmoTH-Coder finetunes CodeLLaMA utilizing 260k program-of-thought data. It is crucial to note that the training data spectrum of MAmmoTH encompasses our training set, incorporating data from the training sets of GSM8K, MATH, and AQuA. To ensure a fair and accurate comparison, we employ the test set of MATH provided by Yue et al. (2023), consisting of $4,097$ samples, which is marginally smaller than the official MATH test set.[4] Additionally, we introduce a variant of SEGO, termed SEGO (-Sequential & Subgoal), which is a version where the policy model is exclusively trained with instruction tuning data, sourced from `gpt-3.5-turbo-0613`. The comparative results, as illustrated in Table 3, reveal that SEGO consistently outperforms MAmmoTH-Coder, across different parameter sizes on both GSM8K and MATH datasets, underscoring the efficacy of the sequential subgoal optimization framework.

## F CASE STUDY

In this section, we delve into a specific example to illustrate the efficacy of our model, depicted in Figure 5. In this figure, the elements labeled as the problem, sub-problem, and solution (of the sub-problem) correspond to the final goal, intermediate goal, and intermediate state, respectively. The sub-problem showcased is derived through the sequential subgoal optimization process. Additionally, we provide the full solution, which is derived from the solution of the sub-problem. This case study indicates the model's capability to search for a suitable sub-problem that ultimately facilitates the derivation of the accurate solution to the final goal.

---

[4]`https://github.com/TIGER-AI-Lab/MAmmoTH/tree/d4dca8947e9382cc8f2b627620e887bec47d3c76/math_eval/dataset/math`

**Problem:** A four-digit code can consist of the digits 0 ~ 9 except that it does not contain the digits 1, 2, and 4 at all. If repeated digits are allowed, what is the probability that it has at least one even digit?

**Sub-problem:** A four-digit code can consist of the digits 0 ~ 9 except that it does not contain the digits 1, 2, and 4 at all. If repeated digits are allowed, what is the probability that it has at least one even digit?
**Solution:**
```
def solve() -> float:
    total_codes = 0  # Total number of possible codes
    codes_with_even_digit = 0  # Number of codes with at least one even digit

    # Iterate through all four-digit codes
    for i in range(10000):
        code = str(i).zfill(4)  # Convert the code to a four-digit string

        # Check if the code contains only digits 0, 3, 5, 6, 7, 8, and 9
        if all(digit in ['0', '3', '5', '6', '7', '8', '9'] for digit in code):
            total_codes += 1

    return total_codes
```

**Full solution:**
```
def solve() -> float:
    total_codes = 0  # Total number of possible codes
    codes_with_even_digit = 0  # Number of codes with at least one even digit

    # Iterate through all four-digit codes
    for i in range(10000):
        code = str(i).zfill(4)  # Convert the code to a four-digit string

        # Check if the code contains only digits 0, 3, 5, 6, 7, 8, and 9
        if all(digit in ['0', '3', '5', '6', '7', '8', '9'] for digit in code):
            total_codes += 1

            # Check if the code has at least one even digit
            if any(digit in ['0', '6', '8'] for digit in code):
                codes_with_even_digit += 1

    # Calculate the probability as the ratio of codes with even digit to total codes
    probability = codes_with_even_digit / total_codes
    return probability
```

Figure 5: A case from the training data.

