# OpenReview forum: "SEGO: Sequential Subgoal Optimization for Mathematical Problem-Solving"
_ICLR.cc/2024/Conference — ICLR 2024 Conference Withdrawn Submission_

### Official Review · Reviewer_A5yj · 2023-10-27

**Soundness:** 3 good
**Presentation:** 2 fair
**Contribution:** 3 good
**Rating:** 6
**Confidence:** 3

**Summary:**

This paper proposes a novel framework called SEquential subGoal Optimization (SEGO) to sequentially refine subgoals. By establishing a connection between the subgoal breakdown process and the probability of solving problems, SEGO identifies better subgoals with theoretical guarantees. By incorporating these optimized subgoals into the policy model training (i.e., finetuning of CodeLLaMA) leads to significant improvements in mathematical reasoning.

Contributions: 1) It highlights the importance of strategic subgoal optimization and proposes a subgoal optimization method with LLM and reinforcement learning; 2) It demonstrates a notable improvement in the mathematical reasoning capabilities when applying the proposed method to finetune CodeLLaMA.

**Strengths:**

(1) The paper introduces the new idea of subgoal optimization in mathematical problem solving, and proposes an innovative approach that combines RL and LLM to realize it.

(2) The paper provides a solid theoretical foundation.

(3) Experimental results demonstrate substantial performance enhancements.

**Weaknesses:**

(1) The paper lacks illustrative examples, making it less accessible for readers to comprehend. It is advisable for the authors to incorporate a specific mathematical problem as an example to achieve the following objectives:

a) Specify some RL concepts (e.g., actions, states, and goals) within the context of mathematical problem solving.

b) Present the specific input and output of the five pivotal modules (i.e., subgoal generator, policy model, etc), when addressing the given example problem.

c) Demonstrate how the initially generated sub-goals are optimized step by step, when addressing the given example problem.

(2) In Figure 1, the relationships between various components are not clearly depicted. For instance, it is unclear how the left part relates to the right part, and what the relationship is between the upper right part and the lower right part. Clarifying these relationships in the figure would improve the visual presentation of the paper.

**Questions:**

Questions:

(1) What is the difference between the training stage and the test stage in SEGO? In the test stage, does only the policy model come into play, meaning you input a problem into the policy model and receive the solution?

(2) In Chapter 4.2, does "N" refer to the number of rounds for sampling subgoals for one problem, and does "η" represent the maximum number of subgoals for one problem?

Suggestions:
See "Weaknesses".

---

### Official Review · Reviewer_PGvU · 2023-10-27

**Soundness:** 2 fair
**Presentation:** 2 fair
**Contribution:** 3 good
**Rating:** 5
**Confidence:** 3

**Summary:**

This paper introduces a novel framework named SEquential subGoal Optimization (SEGO), designed to enhance the problem-solving capabilities of Large Language Models (LLMs) in the context of mathematical tasks. SEGO utilizes five networks to solve the problems through the subgoal breakdown process. The paper also provides theoretical proof for SEGO. In their experiments, SEGO outperforms other LLM models on two benchmarks, GSM8K and MATH.

**Strengths:**

This paper proposes an interesting method to optimize the policy model for use in the context of mathematical tasks, with some explanations and proofs provided. The experiment shows that the proposed method achieves better results.

**Weaknesses:**

The paper is hard to follow. The definitions 1, 2, and 3 are defined without any explanations. These definitions heavily rely on mathematical expressions and formulas in the implementation of the SEGO framework. This confuses the reader and obscures the actual experimental steps. For instance, in section 2.3 Sequential Subgoal Optimization, the subgoal generator, and the subgoal optimizer are defined as LLM. However, the definition uses them like mathematical functions instead of text generators.

**Questions:**

1. How to use SEGO? The author should provide an example from GSM8K and explain it step by step. For example, given a problem, SEGO breaks this problem into more than one subproblem, and includes solutions, and the final answer. Following the case study and Figure 5, it is hard to fully understand how to use the SEGO during solving the problem.
2. The Subgoal Generator, Subgoal optimizer, Likelihood model, et al. are language models (B.5), but in Definitions 1,2, and 3, they are used like mathematical functions instead of text generators. How to use language models' output as a math function? What are these model's actual behavior in the whole framework?
3. In Definition 3, "Let the process start with the sampling of $\omega_{\eta-1}$ from f(·|g, s). Sequentially, $\omega_{\eta-2}$ is derived from $\omega_{\eta-1}$ via the transition operator $T_{\eta-1}$", what is the transition operator actually doing? What is the meaning of the "transition operator" during the different intermediate state and intermediate goal? Is $T$ a word, sentence, or feature? It would be better to provide an example.
4. In Figure 1, the subgoals generated by the subgoal optimizer h, whether accepted or rejected by r&M, lead to the same outcome. What is the difference? In addition, the meaning of the green goal in the lower right corner is not explained.

Other minor issues:
* Paragraph 2, line 6, "The improvement is then assessed by considering feedback from the other modules." What modules?
* Definition 1, for j ∈ {0, 1, . . . , $\eta-1$}, the $\eta$ first appeared in the paper without further explanations. What is the meaning of $\eta$? In the experiments, $\eta$ is defined as "length of sequences". Does this mean the number of tokens in a sentence?
* What is the meaning of $\omega\prime$ in the Definition 2?
* In 4.1 Ablation Study, SFT is not defined.
* In Figure 2, the coordinates of the two figures are not consistent.
* In Figure 4, the coordinates on the left and right sides are not aligned.
* In Algo 1, how to sample $\omega^{i}_{j+1}$ from q(·|$\omega^{i}_{j+1}$, g, s)? What is q()? Is this also LLM?
* In Definition 3, how to ensure that the $\omega_0$ will be sampled by subgoal generator f(·|g, s)?
* The symbol representing the (s,g) in Figure 1 is used by "w", but the definition used $\omega$.
* There are too many mathematical functions in the current Algorithm 1, making it hard to understand. Consider rewriting it.

---

### Official Review · Reviewer_bT9K · 2023-10-30

**Soundness:** 3 good
**Presentation:** 3 good
**Contribution:** 3 good
**Rating:** 6
**Confidence:** 3

**Summary:**

This paper introduces a novel framework called SEGO (SEquential subGoal Optimization) that focuses on the use of subgoals to optimize mathematical problem-solving through Large Language Models (LLMs). Leveraging the power of goal-conditioned Reinforcement Learning, SEGO aims to decompose complex mathematical challenges into manageable subproblems or subgoals. This approach is inspired by the principle that breaking a problem down into subgoals can simplify its solution.


The SEGO framework functions by proposing a draft waypoint (termed subgoal) and optimizing it in a sequential manner. It utilizes a reward model, a subgoal generator, a subgoal optimizer, and a likelihood estimator. These components work in tandem to propose, evaluate, and refine subgoals, ensuring that the system doesn't get trapped in less beneficial ones. The aim is to maximize the probability that a mathematical problem, represented as a goal, can be reached from an initial state. The efficiency of this method is rooted in its ability to quantify the likelihood of reaching a goal based on the selection of subgoals.

**Strengths:**

1 Innovative Framework: The paper introduces SEGO (SEquential subGoal Optimization), a groundbreaking framework that combines goal-conditioned Reinforcement Learning with the potential of Large Language Models (LLMs) for mathematical problem-solving.


2 Theoretical Foundation: SEGO's strength also lies in its solid theoretical underpinnings, which not only describe its functionality but also provide a rationale for its design and expected outcomes.


3 Exceptional Experimental Results: The efficacy of SEGO is rigorously validated through experiments on significant benchmarks like GSM8K and MATH. Not only did it demonstrate superiority over existing methods, but the results also highlighted the practical viability and effectiveness of the SEGO framework in real-world mathematical problem-solving.

**Weaknesses:**

1 Narrow Focus on Subgoals: One of the most pronounced limitations of the SEGO framework is its exclusive concentration on the sequential chain of subgoals. This singular approach may not holistically capture the multifaceted nature of mathematical problem-solving. Human cognition often involves a complex interplay of multiple strategies and not just a linear progression of subgoals.


2 Over-reliance on GPT-3.5 Instructions and Limited Scalability: The experimental outcomes presented in the paper are significantly contingent upon the instructions generated by GPT-3.5. While SEGO demonstrates prowess in the given experiments, its scalability to more complex mathematical challenges (in which GPT-3.5 is not able to generate subgoals) or other domains remains untested.

**Questions:**

1 Could you include the performance of GPT-3.5 in Table 1? Were there instances during experimentation where GPT-3.5 failed to generate appropriate subgoals? If so, how did this impact the overall results? Could you provide a comparative analysis of the evaluation results on datasets where GPT-3.5 was successful in generating correct subgoals versus those where it wasn't?


2 Within the framework's 'chain of thought', can every thought be explicitly labeled as a subgoal, or are there nuances that distinguish certain thoughts from being subgoals?


3 When deploying GPT-3.5 to generate subgoals, is there a way to specify or limit the number of subgoals(Number of sequences) desired for a particular problem? How does the framework handle scenarios where an excessive number of subgoals are generated? I do not find relevant prompts in the appendix.

---

### Official Review · Reviewer_DwRV · 2023-11-01

**Soundness:** 2 fair
**Presentation:** 2 fair
**Contribution:** 2 fair
**Rating:** 3
**Confidence:** 4

**Summary:**

This paper proposes a framework for improving LLM performance when solving math problems. The proposed method breaks each task into subgoals, while attempting to accurately measure the probability that a given subgoal will lead to a solution to the overall problem. The paper claims that a modified version of Expectation Maximization establishes this link in a theoretically-justified and practical way. Then it presents results on math problems from common benchmark problem sets, and compares against several baseline LLMs.

**Strengths:**

Math problems are known to be an area where LLMs perform poorly, so a method that can improve their performance on math would be quite valuable. This paper seems to do that: performance seems to improve substantially over previous methods.

**Weaknesses:**

The details of the method are extremely fuzzy. The paper makes almost no reference at all to how the method connects to LLMs until the appendix, where it finally presents instruction templates for each of the LLM modules, along with an algorithm, and a sample output. Since these don't appear in the paper itself, the actual method remains unclear throughout the entire main text. These do not adequately explain the method, the math-heavy pseudocode is highly non-intuitive, and the sample output presented in Appendix F (which contains an identical goal and subgoal) is insufficient to illustrate how the method is supposed to work. Or if it is indeed representative, then the method does not appear work by the mechanism outlined in the paper.

I strongly recommend that the authors rewrite the paper to incorporate all of Appendices B, C, and F into the main text (preferably expanding on Appx C substantially), as these are crucial for the reader to have any hope of understanding the paper.

I was left wondering what was meant by the terms "states", "actions", "trajectories". It seems like states must be LLM contexts. But then what's the difference between state ("combination of actions taken") and trajectory ("sequence of actions executed")? And what is an example of an action? It seems like the paper is badly in need of a running example illustrating how the method would operate on a particular math problem.

I did not check the math thoroughly, due to lack of understanding of the overall framework. Apparently the paper explains how some of the terms "[steer] the process of picking subgoals", "[offer valuable insights into whether a subgoal is beneficial [...]", and "serve as a corrective mechanism to prevent the agent [from getting stuck]". Unfortunately, it is not clear how these terms accomplish any of these objectives. I still do not understand how "SEGO establishes a connection between the subgoal breakdown process and the probability of solving problems."

The paper appears to extend two 2022 papers, by Drori et al. and Chen et al., which use the "program of thought (PoT)" approach to solve math problems. Unfortunately, the submitted paper never explains what PoT is.

If I had to guess, I'd say the method translates word problems into a state/goal, then a submodel proposes a subgoal, which an LLM then tries to solve by writing a program? And somehow a chain of programs solves the overall goal?

The experimental results seem positive, but without error bars, it's hard to know if they are significant, especially given that all of these models generate outputs stochastically. Furthermore, the ablation study is missing half the possible ablations. What about -Subgoal? -SFT? -Subgoal&SFT?, or -Seq&SFT? These would help determine the relative importance of the different model components.

In section 4.3, the paper claims a "positive correlation between the progression of training steps and the percentage of valid subgoals." How strong is the actual correlation though? It seems very weak.

In the subsequent paragraph on "Hardness of Problems", does the final sentence mean that as training progresses the average problem difficulty with valid subgoals increases?

In section 5.2, it's not always clear how the RL related work is actually related to the work in the paper.

**Questions:**

I don't have major questions for the authors, because my main concern is simply that the writing in the paper is too unclear to accurately describe the method. I have tried to outline the aspects of the paper that I was confused about in the previous section, and I would welcome explanations that shed new light on those areas.